# Simulation Research of Hydraulic Stepper Drive Technology Based on High Speed On/Off Valves and Miniature Plunger Cylinders

**DOI:** 10.3390/mi12040438

**Published:** 2021-04-15

**Authors:** Hanyu Qiu, Qi Su

**Affiliations:** The State Key Laboratory of Fluid Power and Mechatronic Systems, Zhejiang University, Hangzhou 310058, China; qiuhy@zju.edu.cn

**Keywords:** hydraulic stepper drive, high speed on/off valve, miniature plunger cylinders, simulation research

## Abstract

The technology for stepper drive that can achieve accurate motion in the hydraulic field has always been urgently needed in the industry. This paper proposes a hydraulic stepper drive based on five high speed on/off valves and two miniature plunger cylinders. The stepper drive discretizes the continuous flow medium into fixed small steps through the miniature plunger cylinder and realizes the state control of the drive through the logic action of the high speed on/off valve. This paper improves the current stepper drive and establishes a mathematical model to analyze the error of the drive and calculate the position of the actuator. In addition, through simulation research, the performance parameters such as the single-step step characteristic and pressure characteristic of the stepper drive are studied. The results show that, compared with the technology of current stepper drive, this stepper drive can effectively remove the “post step” phenomenon, greatly improve the stepper accuracy of the stepper drive, and have a more excellent performance.

## 1. Introduction

The hydraulic drive for position control has been broadly used in multiple industries for decades due to its high-power density, fast response, and low cost [1,2]. A few applications are machine tools, wheel loaders, or rescue robots [3,4,5]. The closed-loop electrohydraulic proportional or proportional servo technology is widely used at present [6,7,8]. The closed-loop control system consists of an electro-hydraulic servo / proportional control valve, hydraulic cylinder/motor, displacement/angle sensor, and closed-loop controller [9]. The closed-loop control system can provide excellent dynamics and precision if all components have adequate accuracy and dynamic characteristics. However, this technology still has some defects: high costs because of extensive use of sensors, the sensors’ instability caused by the bad working conditions, no power output or even energy loss of the system when the actuator is at the target location [10]. For industry, cost and reliability are the primary considerations. The sensor and its accessories increase the cost and risk of instability of the position control system. Once it is unstable, it may cause severe economic loss or safety accidents. Therefore, the open-loop hydraulic stepper drive technology, which reduces the application of sensors in the system, is urgently needed in the industry.

The related concept of technology for hydraulic stepper drive was proposed as early as the 1940s [11,12]. However, due to the components’ slow response, the system’s dynamic performance is poor. In recent years, with digital hydraulic technology development, various solutions for hydraulic stepper drive appear. Some scholars have proposed a scheme of transforming the actuator. Milecki, A. et al. [13] and Yagci, B. et al. [14] proposed stepper drive cylinders that use the stepper motor to output step control signal in order to drive the hydraulic valve spool. Dindorf, R. et al. proposed a digital cylinder, in which stepper position grooves were set on the hydraulic cylinder, but the stepper position in this scheme is limited [15]. Based on the basic principle, the constant displacement hydraulic pump’s output flow is proportional to the revolutions. Some scholars have proposed a scheme of reforming the pump. Minav, T.A. et al. designed a direct drive position control system for the forklift fork. While lifting the load, the hydraulic pump produces a flow depending on the rotating speed of the servo motor. [16]. Carlobonato et al. designed a stepper drive system, in which the position and speed of the double-acting cylinders in the system depended on the oil inlet flow of the pump [17]. However, due to the leakage of the hydraulic pump, the displacement control error was significant. The others proposed a scheme of using high speed on/off valves instead of traditional valves to control servo-pneumatic systems. For instance, Ahn et al. proposed a novel modified pulse width modulation (PWM) valve pulsing algorithm that employed eight high speed on/off valves to control a rod-less cylinder and the position control was successfully implemented [18]. Marko Agrež et al. presented a novel concept of design guidelines on selecting an appropriate inlet steam turbine control valve assembly and developed a new inlet steam turbine control valve assembly comprising twelve control valves [19]. Gradl et al. proposed a new energy-saving hydraulic stepper drive with a hydraulic plunger cylinder and 3/2-way valve. With the hydraulic steeper drive, the flow rate can be controlled rather directly by adjusting the switching frequency [20]. The high-speed valves are cheap and robust, and they have a rapid response. Therefore, the position control system of high speed on/off valve has been applied on various occasions [21,22,23].

Most of the schemes, which use high speed on/off valves, control the drive’s flow by adjusting the duty cycle of valve. Due to the nonlinear characteristics of fluid systems and influence of environment, the accuracy of these position control schemes needs to be improved. Lots of control methods have been tried by researchers. Najjari et al. proposed a position controller for a double-acting cylinder. The system employed four on/off valves and a classical proportional integral derivative (PID) controller optimized by genetic algorithm [24]. Bruno, N. et al. proposed a novel pneumatic high speed on/off valve driven by a piezoelectric stack actuator. A closed-loop control system based on a combination of PWM signal and a PID controller was designed in order to use the designed high speed on/off valve in the control of a pneumatic cylinder. The results showed that the high speed on/off valve controlled pneumatic cylinder has good position tracking performance [25]. These schemes using PID controllers also need to add sensors, which increases the cost of the system.

Some scholars combine high speed on/off valve with miniature plunger cylinders. The cylinders can limit the single-step position of steeper drive. Nobuhiro et al. proposed and tested a flexible linear stepping actuator using typical pneumatic cylinders as an auxiliary component for the rehabilitation and power assist devices. The position control system using pneumatic driven brakes and seven on/off valves was also proposed. However, the results showed that the downward movement of the actuator is much affected by the gravity of the load mass [26]. Gradl et al. proposed a linear hydraulic steeper drive. The drive is realized by a hydraulic cylinder piston unit which displaces defined fluid quantum by limited forward strokes of that piston [27]. Nevertheless, under the influence of the non-zero dead zone volume, this kind of scheme of the actuator after the main step will produce a tiny step phenomenon, called “post step”. The phenomenon of significant influence on the precision of the stepper drive.

This paper proposes a new type of technology for hydraulic stepper drive without any position sensor and designs a stepper drive. The stepper drive uses five high speed on/off valves and two micro plunger cylinders to form a stepper drive. The control system can discretize the continuous fluid transmission medium into fixed tiny steps, and the position control of the actuator can be completed only by controlling the number of steps. This paper establishes a mathematical model to explain the working principle of the hydraulic stepper drive and explains and analyzes the “post step” phenomenon. Under the action of system pressure and load pressure, the oil in the dead space volume of the micro plunger cylinders and the pipeline expansion. The new stepper drive prevents this part of the oil from entering the actuator by adding a fuel de-livery control valve. Compared with the original stepper drive technology, this technology for stepper drive can remove the “post step” phenomenon, improve the stepper’s accuracy greatly, and have more excellent performance.

## 2. Hydraulic Stepper Drive

The hydraulic stepper drive is based on a simple working principle. It makes use of the precise and a priori has known displacement of an incompressible fluid by an end-to-end motion of a piston in its cylinder. Therefore, the reciprocating motion of the piston can transfer the fixed fluid particles to the actuator hydraulic cylinder through the pipeline. The actuator hydraulic cylinder starts to move step by step. Based on this principle, the widely used scheme of the current hydraulic stepper drive technology is shown in Figure 1. There are various hydraulic circuit designs to realize the hydraulic stepper drive, but the basic functionalities are similar.

When the reset control valve is open, the reset spring pushes the piston of the miniature plunger cylinders to its initial position. During the movement, the actuator hydraulic cylinder ideally performs no position step since the fluid displaced on the bottom of the miniature plunger cylinders is consumed at the top side. However, in real working conditions, the hydraulic stepper drive will be affected by the compressibility of the fluid and the friction of the pipeline. Therefore, during the movement, the fluid of the current hydraulic stepper drive will enter the actuator hydraulic cylinder to make the actuator hydraulic cylinder advance a small step again. This phenomenon will have a great impact on the accuracy of the stepper drive. This paper proposes a new way to add a fuel delivery control valve at the fluid outlet of the hydraulic stepper drive. When the reset control valve is open, the fuel delivery control valve can isolate the actuator hydraulic cylinder from the hydraulic stepper drive in order to prevent the mutual influence from two parts. The schematic of improved hydraulic stepper drive and timing diagram is shown in Figure 2.

As is shown in Figure 2, a completely controllable hydraulic stepper drive includes two miniature plunger cylinders and five high speed on/off valves. The hydraulic stepper drive includes six working modes: step forward mode, step backward mode, fast-forward mode, fast-backward mode, position holding mode, and unloading mode. The mode can be switched through the logic action of the high speed on/off valves. The logical relationship between the state of the stepper drive and the control valves is shown in Table 1.

“Step Forward”: open the fuel supply control valve V1 and close the fuel return control valve V2. The stepper drive is in the step forward mode. At the same time, if the advance and retreat control valve V3 and the fuel delivery control valve V5 are open and the reset control valve V4 is kept closed, the system pressure encourages the fluid to enter the non-spring cavity of the step forward cylinder. The piston in the step forward cylinders starts to move from the bottom. When it moves to the limit position, the step forward cylinder transfers the fluid of the total cavity volume into the actuator through the hydraulic circuit. The actuator hydraulic cylinder moves one step forward, and the step size is the ratio of the volume of the step forward cylinder cavity to the cross-sectional area of the actuator hydraulic cylinder. After completing these actions, close the advance and retreat control valve V3 and close the fuel delivery control valve V5 and open the reset control valve V4. The oil inlet and outlet ports of the step forward cylinder are connected through the reset control valve V4, and the pressure in the cylinder is equal. The spring pushes the piston back to its initial position. The oil in the non-spring cavity of the step forward cylinder enters the spring cavity. The step backward cylinder does not work during the whole process. “Fast forward”: open the fuel supply control valve V1 and close the fuel return control valve V2 at first. And then keep the advance and retreat control valve V3, the reset control valve V4 and the fuel delivery control valve V5 open at the same time, so that the hydraulic oil will directly enter the actuator hydraulic cylinder without using the step forward cylinder.

“Step backward” and “Fast backward”: open the fuel return control valve V2 and close the fuel supply control valve V1. The stepper drive is in the step backward mode. The rest of the working principle is exactly the same as step forward mode. “Position holding”: close the fuel supply control valve V1 and the fuel return control valve V2. The load port of the stepper drive is disconnected from the oil inlet and return port. The pressure and the position in the actuator hydraulic cylinder remain constant. “Unloading”: when the fuel supply control valve V1 and the fuel return control valve V2 are opened at the same time, and the advance and retreat control valve V3 is kept closed, the oil inlet of the stepper drive is connected to the oil return port, and the system is in an unloading state.

The characteristics and advantages of the proposed concept can be summarized as follows:The control system can divide the continuous fluid transmission medium into fixed tiny steps, and the position control can be completed only by controlling the number of steps.This is an open-loop control system, which does not exist such problems: oscillation, instability, and the lack of robustness.The drive has multiple modes to cope with different working conditions.When the actuator stays at the target position, the system is almost no leakage, and the energy consumption is low.The step size can be adjusted by connecting miniature plunger cylinders in parallel or controlling the stroke of the cylinders.The compressibility of hydraulic oil in actual working conditions may affect the step accuracy.The switching state of stepping control components such as high speed on/off valves may cause pressure fluctuations of the transmission medium and actuator vibration.

## 3. Modeling

In Figure 3, the scheme block diagram of the open-loop hydraulic stepper drive for position control system is shown. Its input signal is the number of steps to be done by the actuator. The high speed on/off valves’ logical action makes the piston in miniature plunger cylinders reciprocating. The reciprocating motion of the piston can transfer the fixed fluid particles to the actuator hydraulic cylinder. The actuator hydraulic cylinder starts to move, and the position control is realized.

### 3.1. Basic Theory

The high speed on/off valve is controlled by PWM signals. For one period, the driving signal can be expressed by [28]:(1)$$D=\frac{T_{on}}{T_{p}}$$
(2)$$u_{p}(t)=\{\begin{array}{cc} U & t\leq DT_{p}\\ 0 & t>DT_{p} \end{array}$$
where D denotes the duty cycle of PWM signal, $T_{on}$ denotes pulse width, $T_{p}$ denotes signal period, $u_{p}(t)$ denotes the input voltage signal, and U denotes the amplitude of the PWM signal.

It can be seen from the orifice equation as follows that the flow rate of the valve is affected by pressure. The flow number is [29]:(3)$$\lambda =\frac{h_{d}}{\upsilon }\sqrt{\frac{2\Delta p}{\rho }}$$
(4)$$C_{v}=C_{v\max}\tanh\left(\frac{2\lambda }{\lambda_{c}}\right)$$
where λ denotes the flow number, $h_{d}$ denotes the hydraulic diameter, υ denotes the kinematic viscosity, $\Delta p_{0}$ denotes the pressure drop across valve and ρ denotes the density of hydraulic oil, $C_{v}$ denotes flow coefficient, $C_{v\max}$ denotes maximum orifice flow coefficient, and $\lambda_{c}$ denotes critical flow number at which transfer from laminar to turbulent characteristics occurs.

The mean fluid velocity is:(5)$$V=C_{v}\sqrt{\frac{2\Delta p}{\rho }}$$

The flow rate is then:(6)$$Q=C_{v}A_{v}\sqrt{\frac{2\Delta p}{\rho }}\frac{\rho }{\rho (0)}$$
where $A_{v}$ denotes cross sectional area of valve.

According to Newton’s Second Law, the piston motion can be expressed as the following [20,30]:(7)$$m\ddot{x}=\left(p_{C}-p_{L}\right)A_{1}-cx-F_{0}$$
where x, m, $p_{C}$, $p_{L}$, $A_{1}$, c, and $F_{0}$ denote the piston position, the mass of the piston, the pressure in the chamber, the load pressure, the piston area, the spring rate, and the preload of the spring, respectively.

Pressure build-up equations is given by:(8)$${\dot{p}}_{c}=\frac{E}{V_{0}+A_{1}x}(q_{c}-A_{1}\dot{x})$$
where $p_{c}$, E, $V_{0}$, and $q_{c}$ denote the pressure in the dead volumes, the bulk modulus, the dead volumes, and the flows rates.

In order to accurately predict the position of the actuator hydraulic cylinder, it is necessary to calculate the volume of fluid delivered to the actuator. The ideal step size of the hydraulic stepper drive in a single step is the ratio of the stepping plunger volume to the cross-sectional area of the actuator hydraulic cylinder. However, due to fluid compressibility and thermal expansion, the actual step size of the stepper drive has deviated from the ideal step size.

Density is the mass of matter per unit volume, and density is a function of pressure and temperature. This function can be approximated by the first three terms of a Taylor series [31]:(9)$$\rho \left(p+\Delta p,T+\Delta T\right)=\rho \left(p,T\right)+{\left(\frac{\partial \rho }{\partial P}\right)}_{T}\Delta P+{\left(\frac{\partial \rho }{\partial T}\right)}_{p}\Delta T$$

It can also be expressed as:(10)$$\rho =\rho \left(1+\frac{\Delta p}{E_{T}}-\alpha \Delta T\right)$$
with
(11)$$E_{T}=\rho {\left(\frac{\partial p}{\partial \rho }\right)}_{T}$$
and
(12)$$\alpha =-\frac{1}{\rho }{\left(\frac{\partial \rho }{\partial T}\right)}_{P}$$

$E_{T}$ is known as the isothermal bulk modulus, and α is known as the heat expansion coefficient. In a fast-responding system, the change in temperature is usually negligible [32]. If the Equation (9) is integrated into the current pressure and for any temperature, then for any pressure, there is:(13)$$\rho \left(p+\Delta p,T+\Delta T\right)=\rho \left(p,T\right)\cdot \exp\left(\int_{p}^{p+\Delta p}\frac{dp}{E_{T}}\right)$$

Fluid density varies with the applied pressure, which implies that when a given mass of fluid is submitted to a pressure change, its volume changes. For a closed hydraulic circuit, the mass of the fluid is constant, so there is
(14)$$\frac{d\rho }{\rho }=-\frac{dV}{V}$$

In practical engineering systems, the hydraulic fluid always contains some gas, which is normally air. The gas exists as bubbles or dissolves in fluid. For the sake of completeness, it should be mentioned that additional models are necessary [33,34].

### 3.2. Step Calculation Model

Affected by the compressibility of the fluid, there is an error between the actual step size of the hydraulic stepper drive and the ideal step size. On some occasions, with high accuracy requirements, the error may exceed the allowable value, so the error must be compensated. This section takes the step forward mode as an example to analyze the working process of the stepper drive and establish a calculation model. The model does not consider the influence of dynamics and assumes that the connecting pipeline in the stepper drive is not deformed.

In the step forward mode, the piston of the step forward cylinder starts to move under the action of the difference between the system pressure and the load pressure. The piston transfers oil (mass of M1, volume of V1) to the actuator. Due to the difference between system pressure and load pressure, the volume of this part of the oil will change when it enters the actuator, which makes the error between the theoretical step size and the ideal step size of the actuator hydraulic cylinder. The change of the oil volume in this part is $\Delta V_{1}$.

When the piston of the miniature plunger cylinder moves to the limit position, the oil (mass of M2, volume of V2) between the advance and retreat control valve V3, the reset control valve V4 and the miniature plunger cylinder can be regarded as a closed system, which contains the dead volume of the plunger cylinder and the connecting pipeline. When the reset control valve V4 is open, the closed system is connected to the actuator hydraulic cylinder, and the pressure changes from system pressure to load pressure. At this time, the volume of oil will change and push the actuator forward. The change of the oil volume in this part is $\Delta V_{2}$.

Ideal step size of actuator hydraulic cylinder: The ideal step size is the ratio of the miniature plunger cylinder volume to the cross-sectional area of the actuator hydraulic cylinder, which does not consider the compressibility of the fluid and the load pressure. The ideal step size is the most commonly used parameter in the design or use of stepper drives.
(15)$$\Delta s=\frac{V_{1}}{A}$$

A denotes the cross-sectional area of the actuator hydraulic cylinder.

The theoretical step size of the actuator hydraulic cylinder: considering the compressibility of the oil, the volume expansion of the oil in the stepping plunger will occur under the influence of system pressure and load pressure. The theoretical step size of the actuator hydraulic cylinder is longer than the ideal step size. The theoretical step size is related to the operating conditions of the drive and can be calculated according to different operating conditions.
(16)$$\Delta s_{1}=\frac{V_{1}}{A}+\frac{\Delta V_{1}}{A}$$

The actual step size of the actuator hydraulic cylinder: the oil of the above-mentioned closed system produces volume expansion under the influence of the system pressure and load pressure, which causes the step size of the actuator hydraulic cylinder to increase. The actual step size is affected by the volume of the dead zone and connecting pipeline, which is difficult to calculate in practice.
(17)$$\Delta s_{2}=\frac{V_{1}}{A}+\frac{\Delta V_{1}}{A}+\frac{\Delta V_{2}}{A}$$

The full step can be divided into the “main step” and the “post step”. The “main step” is $\Delta s_{1}$, and the “post step” is
(18)$$\Delta s_{p}=\Delta s_{2}-\Delta s_{1}=\frac{\Delta V_{2}}{A}$$

The actual position of the stepper drive after i steps is:(19)$$s=s_{0}+\sum_{1}^{i}\Delta s_{2}$$

$s_{0}$ is recorded as the initial position of the actuator hydraulic cylinder.

Aiming at the “post step” phenomenon that occurs in stepper drives, this paper proposes a new idea to add the fuel delivery control valve V5 in the oil outlet of the stepper drive. The control signal of the oil outlet control valve is consistent with the advance and retreat control valve V3. When the reset control valve V4 is open, the above-mentioned closed system is isolated from the actuator hydraulic cylinder to prevent oil from entering the actuator hydraulic cylinder during the resetting process of the piston. At this time, the actual position of the actuator hydraulic cylinder after i steps of the stepper drive is:(20)$$s=s_{0}+\sum_{1}^{i}\Delta s_{1}$$

### 3.3. Other Rules

This paper establishes a model to study the main characteristics of the hydraulic stepper drive, and some constraints should be added to improve the model. It is assumed that the flow rate supplied to the drive is large enough, and an overflow valve is used to limit the pressure to 100 bar. A constant force is applied directly to the load end, which makes the load force of the actuator constant during the working process. The switching time of the high speed on/off valve is ≤3.0 ms, and the flow rate is 8 L/min @ 50 bar. To keep the model as simple as possible, friction between the cylinder and piston is neglected. In addition, in order to calculate the step size accurately, some parameters such as the diameter and length of the connecting pipeline are set. Other main parameters are shown in Table 2.

## 4. Simulation Results

### 4.1. Observed Step Characteristics

In this section, based on the model, the performance indicators of the stepper drive before and after the improvement are compared. The drive completes step forward, position holding, and step backward in sequence. The simulation result of the original stepper drive is shown in Figure 4a, while the simulation result of the improved stepper drive is shown in Figure 4b.

According to the simulation results, it can be seen that the stepper drive without the fuel delivery control valve V5 occurs to a slight step when the reset control valve V4 is open. As is described in theory in Chapter 3, this is caused by the difference between system pressure and load pressure. The improved drive with the addition of the oil outlet control valve can achieve the expected function, and the steps are smooth and stable during the working process, and there is no “post step” phenomenon.

This paper aims to explain and analyze the “post step” phenomenon; and propose a solution to ensure that the improved drive will not produce the “post step” phenomenon. Other scholars in this field have carried out relevant experiments [24,25]. This paper set the simulation parameters with reference to the experiment. The experimental result in reference [25] is shown in Figure 5.

Figure 5 shows the measured and estimated of actuator. This experiment results are cited to verify the model, and the results show that the “post step” phenomenon is serious. In reference [25], the solution is to increase the step forward cylinder’s stroke to increase the single step size or compact the valve block to reduce the dead zone volume. But these methods can not get rid of the “post step” phenomenon. The construction of the experimental platform is one of the critical points of future work.

### 4.2. Single-Step Characteristics

As to hydraulic stepper drive, in designing, manufacturing and using, the step size is considered the ideal step size, but in different working conditions, there is an error between the actual step size and the ideal step size. This error may exceed the limit, so the error analysis of the stepper drive is very important.

The main defect of the stepper drive comes from the compressibility of the fluid. The change of the load force during the working process will cause the step characteristic of the stepper drive to change. This article sets different load pressures as 3KN, 5KN, 8KN, 10KN, 12KN, 15KN, 18KN, and 20KN for testing. The simulation result of the original stepping drive’s single-step characteristic is shown in Figure 6.

It can be seen from Figure 6 that a full step of the original stepper drive includes “main step” and “post step.” The “main step” is approximately equal to the theoretical step size. As the load force changes gradually, the “post step” is greatly affected by the load force. The error ranges from 2.23 μm to 6.98 μm, which causes the maximum error between the stepper drive’s actual step size and the ideal step size in a single step of the stepper drive to be as high as 13.96%.

The simulation result of the improved stepper drive is shown in Figure 7. This scheme only contains “main step” and there is no “post step” phenomenon. The error between the improved stepper drive’s actual step size and the ideal step size ranges from 0.38 μm to 1.19 μm, and the maximum error is only 2.38%. The oil in the connecting pipe between the reset switch valve V4 and fuel delivery control valve V5 also has a pressure change, but the pressure change is small enough that the volume change is not large. Compared with the stepping drive before, the maximum error of a single step is reduced by 11.58%, so the step accuracy is greatly improved.

### 4.3. Error Analysis

Analyze the error between different step size of the stepper drive and the ideal step size under different loads, and set the step size at 50 μm, 60 μm, 70 μm, 80 μm, 90 μm, 100 μm, and the load force at 5 KN, 10 KN, 15 KN, 20 KN, 25 KN. The simulation result of the original stepping drive is shown in Figure 8.

It can be seen from Figure 8 that as the step size increases, the error of the stepper drive is reduced; as the load force increases, the error of the stepper drive is reduced. The load force has a great influence on the accuracy of the stepper drive. The error of the stepper drive with step size of a 50 μm is 13.45% under the action of a load force of 5KN, and the error of the stepper drive is 1.85% under the action of a load force of 25KN.

The simulation result of the improved stepper drive is shown in Figure 9. It can be seen from Figure 9 that the error range of the stepper drive is 0.17~2.21% under different working conditions. Therefore, the accuracy of the stepping drive can be greatly improved after the fuel delivery control valve V5 is added. In addition, the error can be reduced by increasing the load force and the stroke of miniature plunger cylinders.

### 4.4. Pressure Fluctuation

The pressure fluctuation in the working process of the stepper drive will cause the jitter of the actuator hydraulic cylinder, so the analysis of its pressure fluctuation is also important. Change the constant force of the load port to a spring-damper system and reduce the switching frequency of the on/off valve. Analyze the pressure fluctuation of the oil inlet A of the actuator hydraulic cylinder and the simulation result is shown in the figure below.

It can be seen from Figure 10 that in the original scheme, pressure fluctuations occur when the advance and retreat control valve V3 or the reset control valve V4 is open. However, the improved scheme has pressure fluctuations only when the advance and retreat control valve V3 is open. This is because the fuel delivery control valve V5 can isolate the actuator hydraulic cylinder from the hydraulic stepper drive when reset control valve V4 is open. The peak pressure of the original plan is 9.38 bar, and the peak of the improved plan is 9.55 bar. Adding the fuel delivery control valve V5 can reduce unnecessary pressure fluctuations and make the actuator hydraulic cylinder move more smoothly.

### 4.5. Response

The response of the stepper drive is related to the switching speed of the on/off valve and the spring rate of the miniature plunger cylinder. A complete step cycle includes the opening of advance and retreat control valve V3, the moving of the miniature plunger cylinder piston, the closing of the advance and retreat control valve V3, the opening of the reset control valve V4, and the resetting of the miniature plunger cylinder piston. In the improved scheme, the control signal of the fuel delivery control valve V5 is consistent with the advance and retreat control valve V3, so the response of the stepper drive will not be affected. In addition, the original drive will produce pressure fluctuations when the advance and retreat control valve V3 and the reset control valve V4 are opened, which will cause the actuator to jitter, and too much jitter is detrimental to the response. The improved drive has no unnecessary jitter and can pursue a higher response.

## 5. Conclusions

This paper proposes a hydraulic stepper drive technology based on a high speed on/off valve group and two miniature plunger cylinders. The stepper drive technology is an open-loop control system that does not require sensors during operation and has the advantages of low cost, good robustness, and low energy consumption.

Affected by the fluid compressibility, there is a “post step” phenomenon in the current stepper drive, which will affect the accuracy of the stepper drive. This paper establishes a model of the stepper drive, calculates the ideal step size, theoretical step size and actual step size of the stepper drive, analyzes the cause of the “post step” phenomenon, and proposes an improvement scheme. The improved scheme results in smooth stepping steps, and there is no “post step” phenomenon. In addition, this paper establishes a model for the stepping drive. The simulation results show that compared with the current stepper drive, the improved stepping drive has a significant increase in accuracy under the same working conditions. For example, under the load force of 3 KN, the maximum error of the drive with a stroke of 50 μm is reduced by 11.58%. And there is no unnecessary pressure fluctuation.

The technical scheme and mathematical model of hydraulic stepping drive proposed in this paper laid the foundation for the follow-up research. The construction of the experimental platform is one of the critical points of the future work. This technology will promote the application of hydraulic stepping drive in the industry and enhance the core competitiveness of hydraulic transmission.

## Figures and Tables

**Figure 1 micromachines-12-00438-f001:**
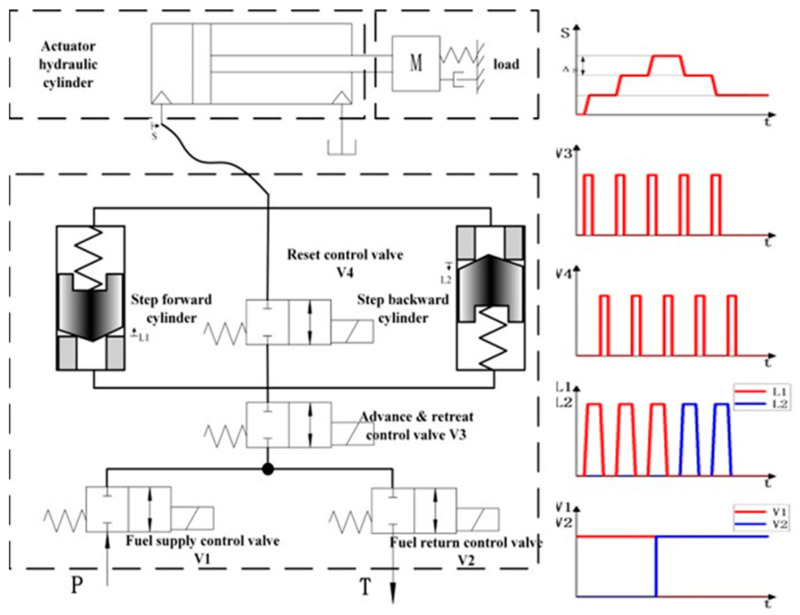
Schematic of the current hydraulic stepper drive and timing diagram.

**Figure 2 micromachines-12-00438-f002:**
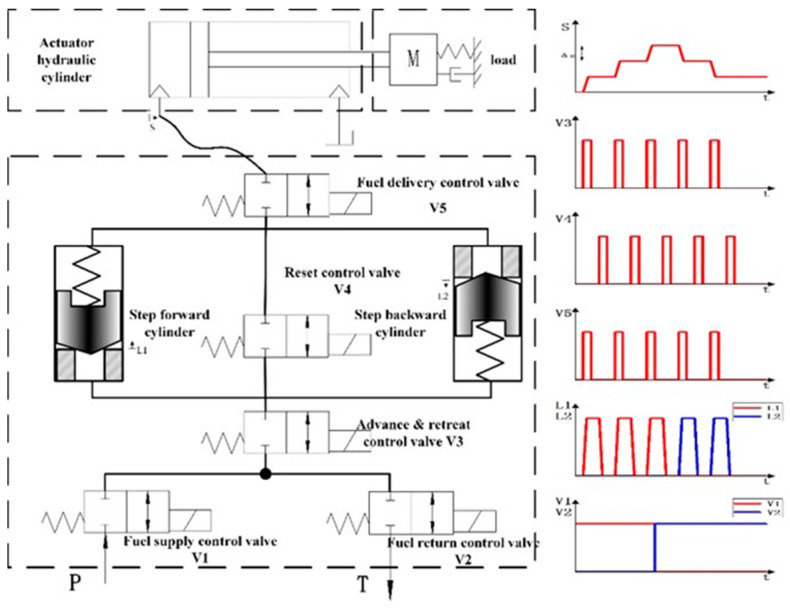
Schematic of the improved hydraulic stepper drive and timing diagram.

**Figure 3 micromachines-12-00438-f003:**
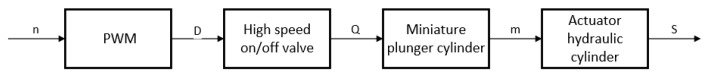
The scheme block diagram of the open-loop hydraulic stepper drive for position control system.

**Figure 4 micromachines-12-00438-f004:**
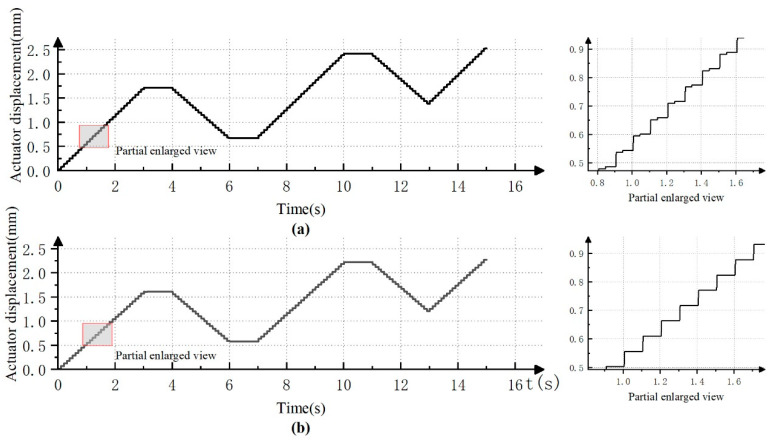
Simulation results of function test of hydraulic stepper drive: (**a**) the simulation result of the original stepper drive; (**b**) the simulation result of the improved stepper drive.

**Figure 5 micromachines-12-00438-f005:**
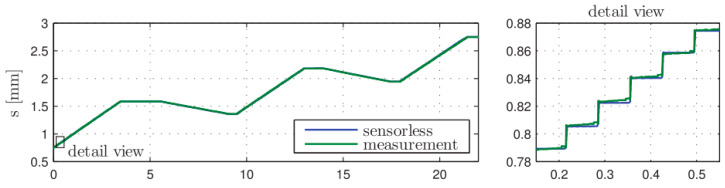
Experimental results of stepping drive in reference [25].

**Figure 6 micromachines-12-00438-f006:**
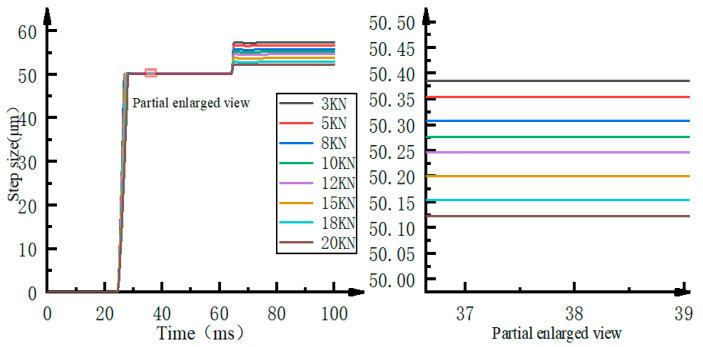
Single-step characteristics of the original stepper drive.

**Figure 7 micromachines-12-00438-f007:**
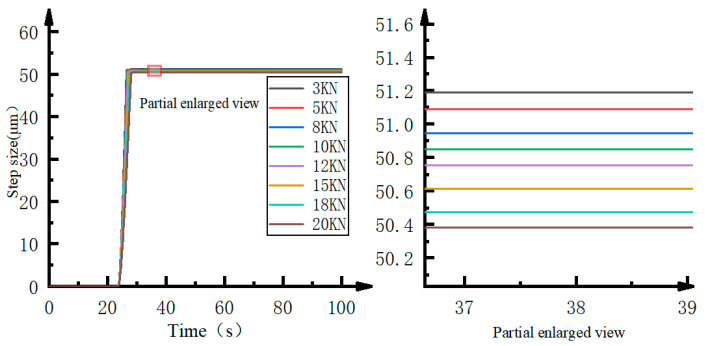
Single-step step characteristics of the improved stepper drive.

**Figure 8 micromachines-12-00438-f008:**
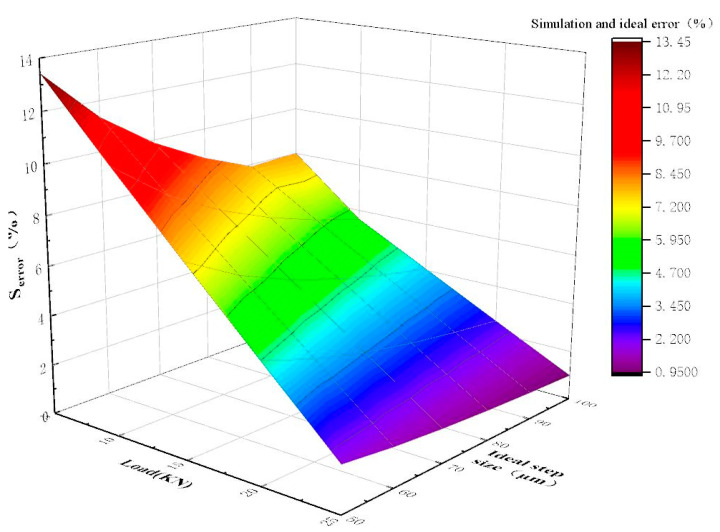
The error between the actual step size and the ideal step size under different working conditions of the original stepper drive.

**Figure 9 micromachines-12-00438-f009:**
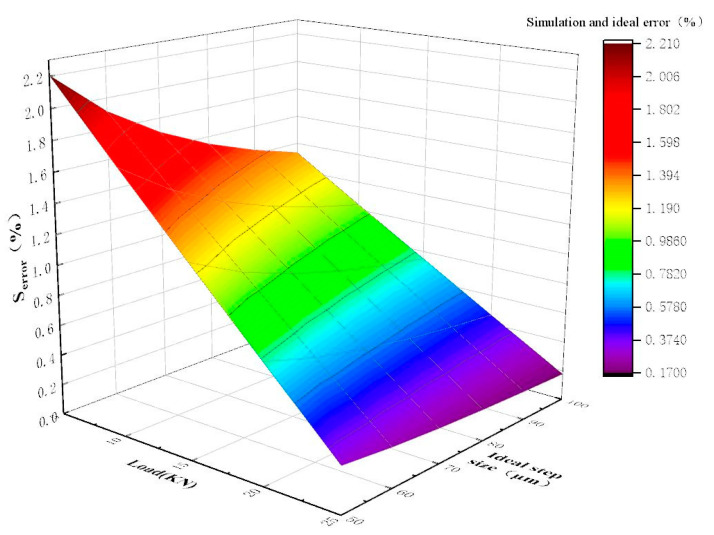
The error between the actual step size and the ideal step size under different working conditions of the improved stepper drive.

**Figure 10 micromachines-12-00438-f010:**
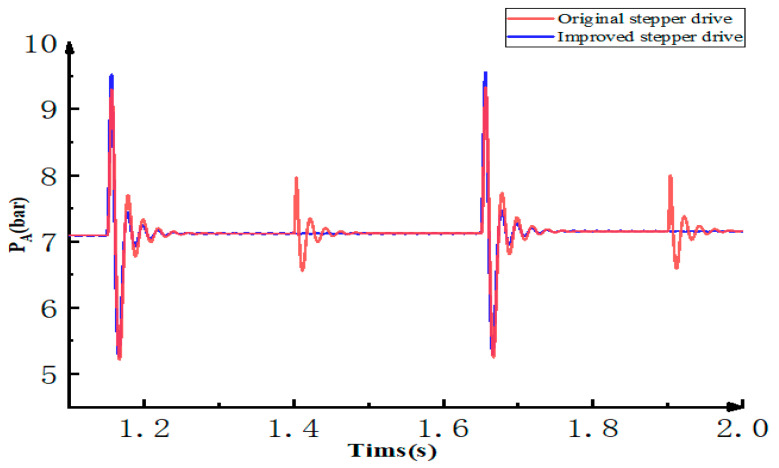
The pressure fluctuation of the oil inlet A of the actuator hydraulic cylinder.

**Table 1 micromachines-12-00438-t001:** The relationship between the state of high speed on/off valves and the function of the hydraulic stepper drive.

Fuel Supply Control Valve V1	Fuel Return Control Valve V2	Advance and Retreat Control Valve V3	Reset Control Valve V4	Fuel Delivery Control Valve V5	Functional Description	
On	Off	On	Off	On	One step forward	Step forward
		Off	On	Off	Piston reset	
		On	On	On	Fast-forward	
		Off	Off	Off	Position holding	
Off	On	On	Off	On	One step backward	Step backward
		Off	On	Off	Piston reset	
		On	On	On	Fast-backward	
		Off	Off	Off	Position holding	
Off	Off	On/Off	On/Off	On/Off	Position holding	
On	On	Off	On/Off	Off	Unload	
		On	On/Off	On	Not allowed	

**Table 2 micromachines-12-00438-t002:** Criterion and constraints for the hydraulic stepper drive.

Parameter	Value
Piston diameter of actuator cylinder r $d_{p}$	60 mm
Rod diameter of actuator cylinder $d_{r}$	30 mm
Ideal step size Δs	50 μm
Piston diameter of miniature cylinder $d_{c}$	10 mm
Length of stroke L	1.8 mm
miniature cylinder dead zone volume $V_{d}$	10 mm^3^
Spring rate of miniature cylinder c	10 N/mm
Diameter of connecting pipe $d_{p}$	5 mm
Length of connecting pipe $L_{p}$	20~50 mm

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
