# Peer review of "Simulation Research of Hydraulic Stepper Drive Technology Based on High Speed On/Off Valves and Miniature Plunger Cylinders"

_micromachines, 2021, doi:10.3390/mi12040438_

Round 1

Reviewer 1 Report

This paper proposed a hydraulic stepper drive technology based on a high-speed on-off valve group and miniature plunger cylinders. This work is lack of novelty. Here are my comments and the authors should address these main issues.  

(1) The review on previous research has some serious gaps. Many recent publications  on have not been touched upon. There is a need to do a comprehensive research review and introduction needs to be rewritten to fill those gaps.

(2)The authors provide some useful information regarding this research but validation.

Author Response

Dear Editor, Dear reviewers

Thank you very much for your valuable comments and suggestions, which are of great help to improve the quality of this paper. We have modified the paper according to your comments, and we believe the suggestions are valuable and enabled us to clarify the analysis and improve the quality of the paper. The detailed modifications in the revision are listed below point by point. The questions and comments are marked in black, and our responses are marked in red.

Thank you for the kind and valuable advice.

With my best regards,

The authors

Reviewer 2 Report

Thank you for inviting me to review the manuscript below:

Manuscript ID micromachines-1183138

Title: Simulation Research of Hydraulic Stepper Drive Technology Based on High Speed On-off Valves and Miniature Plunger Cylinders

This paper focuses on simulation research of hydraulic stepper drive technology based on high-speed on-off valves and miniature plunger cylinders. The concept is interesting, the methodology is well presented but the paper needs some minor revisions. Paper may be publishable as a research paper after the above-mentioned revisions.

  1. The novelty, originality shall be further justified in an introduction that the manuscript contains sufficient contributions to the new body of knowledge. The knowledge gap needs to be clearly addressed.
  2. Describe the current work by other authors in this field. At the end of the introduction add an article structure.
  3. In the modelling section please add the model diagram or model scheme and modelling validation results.
  4. All already known equations must be cited.
  5. The results and discussion should be discussed more extensively by increasing the number of literature. Please compare results by other authors. This article should be cited, https://doi.org/10.1016/j.ijheatmasstransfer.2020.119897.
  6. The manuscript should be read carefully and the English language needs to be corrected.

Author Response

(The authors gave the same response as above.)

Round 2

Reviewer 1 Report

The authors addressed all of my comments and suggestions in the revised manuscript .

Therefore, I suggest this revised manuscript can be accepted for publication.